# Global Changes in Cultivated Area and Breeding Activities of Durum Wheat from 1800 to Date: A Historical Review

Fernando Martínez-Moreno [1,*], Karim Ammar [2] and Ignacio Solís [1]

1   Agronomy Department, Technical School of Agricultural Engineering, University of Seville, 41013 Seville, Spain; isolis@us.es
2   CIMMYT (International Maize and Wheat Improvement Center), Carretera México-Veracruz, Km. 45, El Batán, Texcoco 56237, Mexico; k.ammar@cgiar.org
*   Correspondence: fernan@us.es

**Abstract:** Durum wheat is grown globally on 13.5 million ha in 2020/2021, which amounts to 6.2% of the wheat area. It is assumed that in the past it was more important, but the extent of that importance is unknown. In this work, a historical estimation of the durum wheat area globally was carried out, based on data of the main cultivating countries. Many of the data from the earliest period were based on percentage to all wheat. During the nineteenth century, the percentage of durum wheat to all wheat globally was around 14–16%. However, throughout the 19th and beginning of the 20th century, in America (USA, Canada, Argentina), Asia (Russia, China, India), and Australia, new land was sown with bread wheat, and therefore the percentage of durum wheat fell steadily to 7–9% from 1950 to 2005, and to 6–7% since then. For many years, Russia was the country with more durum wheat cultivation, with around six million ha in the period 1910–1940. Turkey, Italy, Algeria, and India were also big historical players regarding cultivation of this crop. Currently, Canada, Algeria, Italy, and Turkey have the largest durum wheat acreage. The main breeding activities and the future of durum wheat are discussed.

**Keywords:** *Triticum turgidum*; wheat breeding; history of crops; agricultural history





## 1. Introduction

Durum wheat (DW) (*Triticum turgidum* L. subsp. *durum* (Desf.) Husn.) is cultivated on 13.5 million ha (Mha) with a global production of 33.8 million tons in 2020/21 [1,2]. While globally minor, accounting for less than 7% of the total wheat produced worldwide, it is concentrated in relatively small geographic regions where it can be considered as a main cereal crop, contributing significantly to food production and agricultural income. Historically, it is well adapted to low and/or variable rainfall environments, with frequent terminal heat stress, such as the Mediterranean Basin. In fact, countries of the Mediterranean Basin (Algeria, Turkey, Italy, Morocco, Syria, Tunisia, France, Spain, and Greece) account for approximately 50% of world acreage and production. Outside of the Mediterranean region, Canada, Mexico, the USA, Russia, Kazakhstan, Azerbaijan, and India are large to significant DW producers, with the first three being the most important DW exporters. DW is consumed in the form of a variety of products. Pasta products in their numerous form (spaghetti, macaroni, etc.) are by far the most widely produced and industrialized end-products. However, at least one fourth of the global durum production is used in a variety of staple foods such as couscous, bulgur, frike, and different types of flat breads and sweets, some widely industrialized while other produced mostly in the household [3,4]. Adaptation to the highly variable growing conditions of the different Mediterranean micro-environments and quality profiles suitable for the production of local food allowed durum wheat landraces to outcompete their bread wheat (BW) counterparts in the region [5].

However, the DW acreage has been distinguished from that of other wheat species only recently, and even nowadays, statistics on DW cultivated area are not easily accessible for many countries. The website on crop statistics worldwide FAOSTAT does not allow the specification of which type of wheat one is interested in [6]. Most historical statistics refer to all cultivated species of wheat, i.e., BW, DW, and several other species of minor importance (einkorn, emmer, rivet, spelt, Timopheev, Khorasan, etc.). A study on the historical global DW area has never been carried out. The objective of this work was to explore the historical evolution of the DW area of the most important DW growing countries from 1800 to date, deducting from this analysis the evolution of the world DW area, and the percentage of DW within the total wheat area. Major varietal changes over time and main breeding impacts in each country are also included.

## 2. Materials and Methods

Several sources were consulted to get information about the historical DW acreage.

1.  Literature reviews. The books 'Durum wheat breeding' (Royo et al., 2005) [7] and 'World wheat book' (Volumes 1–3) (Bonjean and Angus, 2001; Bonjean et al., 2011, 2016) [8–10] were especially useful for information mining. In many cases, DW acreage is not mentioned as such, but a percentage from total wheat is listed. While generally useful, those percentages resulted in some cases in overestimating the actual DW acreage [11,12]. From 1998 onwards, the data used originated from IGC (International Grain Council, London, UK) Secretariat [1].
2.  Personal communication of DW experts of different countries where official statistics are not always available.
3.  Several historical periods were distinguished in the present work. In the first ones, namely, 1800, 1850, and 1870, the DW area was difficult to obtain and was generally estimated from percentages of the total wheat area. Starting in 1890, the area was reported in a period of 10 years, and, from 2000 on, in periods of 5 years. The year 1935 was included instead of 1940 due to the lack of reliable statistics from involved in World War II. The year 1961 replaced 1960 because it marked the first year that statistics on crop acreage were published by FAOSTAT [6].

Many geopolitical issues had to be taken into consideration in the preparation of this manuscript. Several countries became independent from empires after World Wars I and II, others had modified borders and, finally, the USSR dissolution occurred. In the cases of Russia-USSR, the Austro-Hungarian empire and the greater India, statistics on wheat area corresponding to the years of the existence of the empire/geopolitical entity of that time were recorded. When the geopolitical entity ceased to exist as such, statistics of the newly formed countries were recorded. More importantly, the wheat acreage of Russia (and the former USSR) was not available in many reports of the world wheat situation in the period 1900–1950. The term 'world ex-Russia' was frequent in the statistics of those reports [13–16].

To carry out the study, three tables were prepared (Supplementary Materials). In Table S1, the historical (all) wheat area of 37 important countries over several seasons of the period 1800–2019 was worked out. This table was used as a basis to calculate the DW area when a percentage to all wheat area was provided. The percentage of DW area in every country and season could also be estimated. The world wheat area was also added to this table. The data based on estimations (earlier periods) are in green-colored cells when the total wheat area was also estimated, and in blue-colored when at least a source of information on the total wheat area of a country in a season was available. The most reliable data on wheat and DW area are in orange-colored cells (many were official data, especially after 1961 by FAOSTAT). In some cells, the letter 'n' signified that the country that did not officially exist at that time (e.g., Azerbaijan in 1950). Table S2 includes the DW historical area in 24 countries with significant DW area (Afghanistan, Algeria, Canada, China, Egypt, Ethiopia, France, Greece, India, Pakistan, Iran, Iraq, Italy, Mexico, Morocco, Russia, Khazakhstan, Ukraine, Azerbaijan, Spain, Syria, Tunisia, Turkey, and the USA).

The DW area of Khazakhstan, Ukraine, and Azerbaijan was included in that of Russia during the 19th century and the Soviet period. The DW area of Pakistan before 1947 was included in that of India. The sum of DW areas of those 24 countries plus 3% of the area of the remaining countries was estimated as the DW world area. The additional 3% of the remaining countries was consistent with the known percentage in the period 1998–2020 [1]. Table S3 refers to the percentage of DW to all wheat area (%) of the main historically cultivating countries. Figure S1 shows a video of an animated bar chart of the DW acreage evolution of the 15 main historically cultivating countries in 1800–2020.

## 3. Results

The global historical DW area in the different periods from the 19th century and beginning of the 20th century is provided in Table 1. Data from the 19th century are estimates based on cultivated area percentages from the beginning of the 20th century, but they can be considered as close approximation to the historical acreage of this crop. In the last two hundred years, DW area ranged between 8.4 million ha in 1800 and 19.7 Mha in 1930, with the most recent value of 13.5 Mha in 2020. The peak was reached during the 1910–1930 period with a DW area approaching the 20 Mha. From 1930 to 1950, about two Mha of DW grown worldwide were lost (from 18 to 16 Mha), but the acreage recovered to 18 Mha in the 1970s and 1980s. From the 1980s to 2000, 3 Mha were taken out of DW production, and another 2 Mha from 2000 to 2020. This relatively stable DW area contrasts with the huge growth of the total wheat area, which doubled from 1800 to 1900 (from 55 to 110 Mha). During the 20th century, the wheat area continued to increase, in order to surpass the 200 Mha in 1961. In the 1980s and 1990s, the global wheat area reached 230 Mha and then stabilized at about 220 Mha, the area of today. The percentage of DW to all wheat declined over the studied period. From 1800 to 1900, DW percentage decreased from 16.3% to 13.5%. This percentage first dropped below 10% (to 8.6%) by 1950, and, by 1990, it was 7%. Currently, DW represents 6.2% to all wheat. This percentage drop is mainly attributed to the increase in total wheat area over the period studied.

**Table 1.** Historical durum wheat area and percentage from total wheat area.

| Year [1] | World Durum Area (Mha) | World (Total) Wheat Area (Mha) | Durum Wheat Area (%) |
|---|---|---|---|
| 1800 | 8.41 | 51.5 | 16.3 |
| 1850 | 9.66 | 66.3 | 14.6 |
| 1870 | 11.18 | 84.6 | 13.2 |
| 1890 | 12.84 | 99.9 | 12.8 |
| 1900 | 14.89 | 110.0 | 13.5 |
| 1910 | 17.74 | 132.0 | 13.4 |
| 1920 | 19.60 | 149.5 | 13.1 |
| 1930 | 19.72 | 154.2 | 12.8 |
| 1935 | 18.40 | 157.0 | 11.7 |
| 1940 | 18.66 | 140.1 | 13.3 |
| 1950 | 16.33 | 189.4 | 8.6 |
| 1961 | 16.75 | 204.2 | 8.2 |
| 1970 | 18.78 | 208.0 | 9.0 |
| 1980 | 18.97 | 237.3 | 8.0 |
| 1990 | 16.04 | 230.8 | 7.0 |
| 2000 | 15.88 | 214.9 | 7.4 |
| 2005 | 15.88 | 221.7 | 7.2 |
| 2010 | 13.69 | 215.6 | 6.4 |
| 2015 | 14.29 | 223.4 | 6.4 |
| 2020 | 13.52 | 219.0 | 6.2 |

[1] Data of total wheat area from 1961 were taken from FAOSTAT [6].

The DW historical acreage and percentage to all wheat of the nine main DW-producing countries are presented in Figures 1 and 2. Details on the evolution and area of these

countries (plus Spain and Ethiopia, which do not appear in the figures to facilitate a better visualization of them) are provided in the Section 4. The DW area of Russia steadily increased along the 19th century from a bit more than 1 Mha in 1800 to more than 5 Mha in 1900–1930 (Figure 1). At that time, that acreage accounted for a third of total DW globally cultivated. The area even surpassed 6 Mha in several seasons of the 1920s. After World War II, DW acreage in Russia progressively decreased to about 0.5 Mha, which is approximately the area of today. Turkey had a stable area surpassing 2 Mha until 1970, when it decreased to about 1.5 Mha in the 1980s, keeping that acreage from then on. Italy progressively increased its DW area from 1 to 1.5 Mha until the 1990s and then the area stabilized around the last figure. India has a considerable DW area (ranging between 1 and 2 Mha) until the 1940s when its acreage decreased to around 0.5 Mha, which is the current area. Algeria maintained a stable DW area over 1 Mha, which only fell in the 1990s, but recovered in the 2000s. The DW acreage of USA ranged for many seasons of the studied period between 0.5 and 1 Mha, except for two peaks: one in the 1920s and 1930s, and another in the 1980s (both with around 2 Mha). DW acreage of Morocco, Tunisia, and Syria remain fairly stable. However, the area of Syria decreased from 2005.

Regarding the DW percentage to all wheat, the trend towards a slight decrease was recorded in almost all countries, except Italy (Figure 2). Russia had about 20% of DW acreage to all wheat until the 1940s when it steadily dropped to 3% in 1970, a percentage that has remained approximately the same until today. In Turkey, the percentage fell from 60% to 15% between 1950 and 1970. Then, it stabilized around 15%. Italy is the only country where the DW percentage has increased throughout the period studied, from 20% of the 19th century to 70% of today. Algeria had a large DW percentage close to 95% until the 1960s when it dropped to 67%. Then, it rose again to almost 90%, which is the percentage of today. India maintained a DW percentage of 15% to all wheat until 1940, then it fell to 3% in 1950, which has been maintained until today. In Morocco, a percentage of 70% was maintained until 1980, when it began to drop to 37% in 2000, and is 30% currently. The USA has kept a percentage ranging 4–5% in most of the studied period, except in the peaks of the 1920s–1930s and the 1980s, when it reached 7%. Tunisia has maintained in the studied period a high DW percentage ranging from 80–90%, whereas Syria kept a 70–80%, except in 2005–2015.

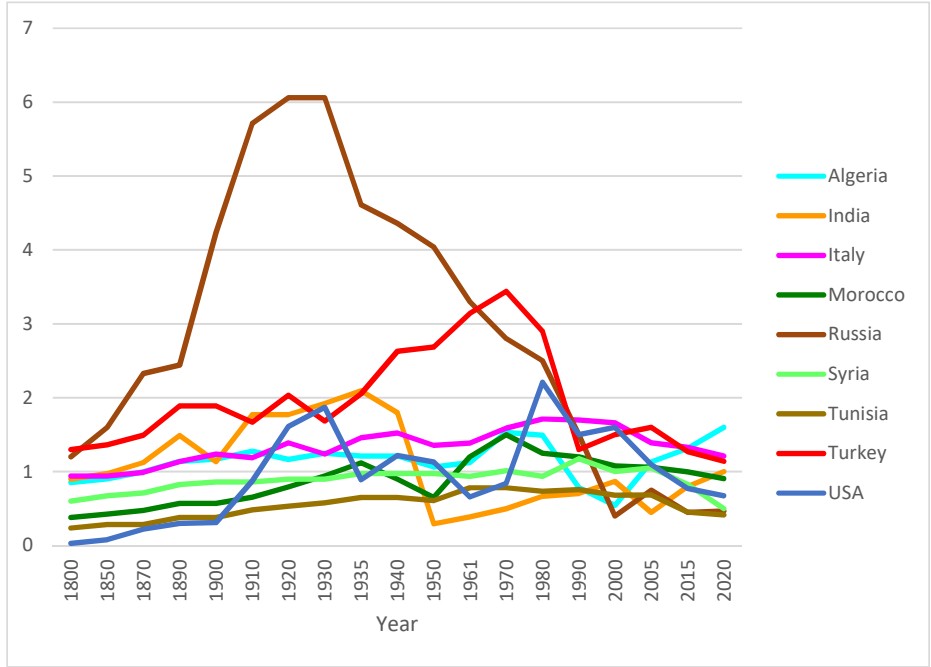

**Figure 1.** Historical area (Mha) in the nine main durum wheat cultivating countries in 1800–2020. Sources of information: [12,13,15,17–32].

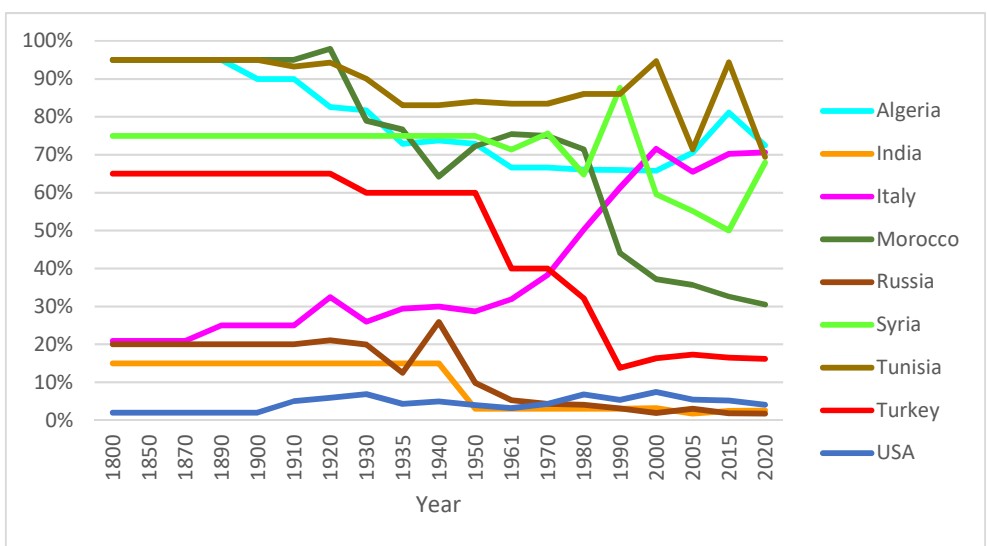

**Figure 2.** Historical durum wheat area (%) to all wheat area in the nine main durum wheat-growing countries in 1800–2020. Sources of information: [14,15,17,19,20,23,29–31,33–48].

## 4. Discussion

### 4.1. Historical Global Durum Wheat Area

Historically, DW was cultivated in regions with mild climatic conditions, such as the Mediterranean Basin and south Black Sea Basin (with spring landraces, planting in late fall), while BW was cultivated in colder temperate zones of Europe and Asia (with winter landraces, planting in the fall) [17]. However, throughout the 18th and 19th century, there was a huge expansion of wheat cultivation by European settlers of new lands in America, Asia, and Australia [29]. Between 1850 and 1950, large areas in Russia, China, and the USA were put into wheat cultivation, and, in 1890–1920, wheat area growth was observed in Argentina, Australia, and Canada. The majority of this new wheat land was sown with BW [29].

The growth of DW area was much lower and mainly attributable to an increase in the cultivation of this crop in Russia in 1870–1920. From 1930 to 1950, a decrease in the DW acreage was observed, which was associated with the decline in cultivation of this crop in Russia [30,31]. During the period 1930–1970, total wheat acreage expanded by about 80 million ha globally, but most of this expansion again involved BW [29]. This unequal expansion of the two crops is probably due, to a large extent, to the industrialization of markets that favored BW for its more suitable gluten characteristics in the industrial production of bread, with DW wheat beginning to be targeted primarily to pasta production, a much smaller market than bread. Another reason for this unequal growth may be related to the expansion of wheat into more difficult environments, those requiring high cold tolerance, winter hardiness, or tolerance to soil toxicities, where BW, thanks to the enhanced variability and adaptability provided by its D-genome (absent in DW), could be more competitive than DW [49]. As a probable consequence of these drivers, a higher investment in technology and intensification was devoted to BW, especially in terms of breeding. Since the late 18th and beginning of the 20th century, many breeding programs were initiated in the UK and France [28,46]. Soon Italy, USA, and Australia followed [47,48,50,51]. Only some Italian breeding programs focused, and partially at that, on DW. The rest of the programs dealt primarily with BW breeding, with a substantial emphasis on winter BW, with breeding for winter hardiness and cold tolerance resulting in the expansion the crop in new cold environments. In the 1960s, new BW spring cultivars were starting to be released by the breeding program, which would later become CIMMYT (Centro Internacional de Mejoramiento de Maíz y Trigo) or the International Center for Maize and Wheat Improvement, which were semi-dwarf, resistant to lodging, resistant to diseases

(especially stem and leaf rust), and highly responsive to fertilizers that became very popular among farmers worldwide [52], further making BW more attractive to farmers than DW. India and the former USSR are examples of this preferential expansion of BW in that period [12,30].

Based on all the information reviewed, an estimate of the proportion of DW area out of the total wheat area of 14–16% in the period 1800–1850 is realistic. This proportion decreased as total wheat acreage expanded. From the late 1960s to 1980s, a modest recovery in the DW acreage was recorded, with the percentage of DW ranging between 17 and 18 million ha. The reasons for this recovery are the following:

1.  New DW cultivars with traits that made them more competitive with BW (dwarfing, resistant to diseases, high yields, wide adaptation) were distributed globally by CIM-MYT to many developing countries with expanding agricultures [53], and others bred by several public institutions and companies in Italy (e.g., Stazione di Granicoltura per la Sicilia) [54].
2.  A growing demand for DW products, especially pasta (globally a trendy food product from the 1970s), but also couscous, bulgur, and frike, in the West Asia and North Africa (WANA) region, which are staple foods.
3.  Generally higher price of DW compared to BW in most markets. Price disparity, almost always favoring DW, is variable, generally 20% higher than the price of BW grain. In countries where DW and BW can be interchanged and markets are relatively free, either crop can be favored in any given year based on the price differential for that particular year. It should be noted that DW commercialization is more restricted due to the smaller number of end-use products, compared to BW, for which it is used [55] and the extent of household consumption.
4.  Policies encouraging farmers to grow DW. Those policies were applied in the European Union, Canada, the USA, and even in North Africa [56].

### 4.2. Cultivation and Breeding in the Main DW-Growing Countries: A Summarized History

Although Canada is currently the country with most DW area in the world, historically the countries with the largest cultivated area were Russia, Turkey, Italy, Algeria, and India. Other important producers from 1800 to present were Morocco, Syria, Tunisia, the USA, Spain, and Ethiopia (see Figure S1).

### 4.2.1. Russia

Historically, rye has been the most important cereal in Russia [57]. However, wheat also played an important role, especially in latitudes below 50–52°, mainly with spring wheat varieties that do not require any vernalization, sown in spring after the snow melt (79.2% from total wheat in 1883–1914, 68.4% in 1917–1990) [31,57]. During the 19th century, DW accompanied Russian settlers into the fertile but semiarid steppe north of the Caucasus and into western Siberia. By 1900, DW had expanded further to the east, being sown along the southern margin of the spring wheat belt, in nitrogen rich soils, and in a warm and dry climate. The presence of railway in Siberia permitted the transportation and commercialization of grain outside the region. DW landraces such as Arnautka, Kubanka, Chernokoloska, Garnovka, and Beloturka from the Caucasus and Volga region (many of them of Turkish origin [58]) were brought to the east [59]. DW was mainly cultivated in four regions (from west to east): Lower Don (north of Black Sea), Middle Volga (north of Caspian Sea), southern Urals, and Altai Krai [30]. A great share of the DW production in the European side of Russia was exported through the port of Taganrog on the Azov Sea, to Italy and France. Taganrog wheats, in fact, a mixture of different landraces from the Don and Volga regions, were famous for their pasta quality in Italy and France [31,60]. The production east of the Volga region was mostly destined to domestic consumption, especially in the form of bread. There was a small share of winter DW planted in regions with relatively mild winters (to avoid winterkill), such as the steppes of Ukraine and the Caucasus region of Russia [42]. Before 1914, the DW area was approximately 6 million ha

(about a third of DW acreage globally), with an annual production of about 4.7 million tons [61]. After the 1917 Revolution, the DW area starts to slowly decrease, especially in the eastern regions. In the period 1920–1940, DW acreage fell from 28% of the spring wheat area (roughly 6 million ha) to 17% (around 4 million ha). In the Kostanai oblast (today northern Kazakhstan), DW dropped from 13.1% of the spring wheat acreage in 1939 to 4.1% in 1952 [30]. After the World War II, DW area continues to recede, being replaced by BW, with more yield, more yield stability, and better response to intensive cultivation. BW breeding programs were successful and yielded cultivars such as Saratovskaya-29 (spring BW), released in 1956 at the Station of Saratov (lower Volga region), and Bezostaya-1 (winter BW), released in 1959 at the Krasnodar Research Institute of Agriculture [31]. While DW breeding efforts existed, they were of lesser resources. The DW cultivar Melyanopus was selected in 1929 from a landrace of the Saratov region, and the cultivar Bezenchukskaya 139 was released in 1980. This latter cultivar was sown for several years on close to 1.4 million ha annually in the former USSR [31]. After the dissolution of the USSR, DW acreage has continued to decrease in Russia, and is currently maintained at approximately 0.5 Mha. Kazakhstan has a DW acreage of about 0.4 Mha, in the north of the country (Akmola and Kostanay regions) and with a tendency towards growth [43].

### 4.2.2. Turkey

DW in Turkey represented around 65–70% of the total wheat area during the 19th century and the beginning of the 20th century (during the Ottoman Empire), which accounted for an acreage of 1.6–2 million ha, making Turkey the second largest DW-growing country after Russia. During the periods that have been looked at, Turkey was always among the world's top three countries in terms of DW area (except in 1930), reaching the top spot during several seasons in the 1970s. DW took up 80% of the wheat area in Southeast Anatolia, 60% in Thrace (European Turkey), and 40% in the coastal areas. In the central Anatolia Plateau where DW occupied 70% of the total wheat area, most landraces were of winter or facultative type [44]. A great diversity of DW landraces, and even wheat species (e.g., einkorn, emmer), was a feature of the Turkish wheat landscape, which is consisted with a long history of these species in the country. The presence of the DW ancestor, the wild emmer in the southeast (Karacadağ of Diyarbakır province), also contributed to this diversity [62]. DW landraces such as Kunduru, Üveyik, Sarı Bursa, Koca Bugday, Bagacak, Sorgul, Beyaziye, Havrani, İskenderi, Karakilcik, Akbasak, and Kibris bugdayi are only some of the many sown by Turkish farmers. The total wheat area expanded from 4.4 million ha to 7 million ha from 1950 to 1960, and to 8.6 million ha by 1970. However, as in many other countries, the expansion involved mostly bread wheat, with the "Green Revolution" spring cultivars originating from CIMMYT, such as Penjamo-62, Mentana, Sonora 64, Lerma Rojo 64, and others from different origins such as Lancer and Bezostaya-1. The main DW national breeding programs are: CRIFC in Ankara, BDMIKHAM in Konya, ANADOLU in Eskisehir, EGE Tar. Ars. Ens. in İzmir, Doğu Akdeniz Tar. Ars. Ens. in Adana, and GAP UTAEM in Diyarbakır. They were assisted by CIMMYT, ICARDA (International Center for Agricultural Research in the Dry Areas), and OSU (Oregon State University) for germplasm obtention and training. Cultivars such as Kiziltan 91, Emin bey, Ege-88, Solen-2002, Fırat-93, Saricanak-98, Zuhre, Ayzer, Sarıbasak, and Kunduru 1149, were developed and grown in recent years. DW cultivars still have a lower production compared to BW, are less tolerant to cold, and many are Zn sensitive (Zn deficiency is a problem in many soils of Turkey) [44]. Currently, DW area ranges between 1.1 and 1.2 million ha, around 16% of the total national wheat area, and nearly 10% of DW acreage globally [1,2].

### 4.2.3. Italy

During Roman times, DW replaced emmer as the main wheat species along the Mediterranean Basin. Wheat was named *triticum* in Latin, while the proper name of DW was robus. However, Romans commonly called DW *triticum*, their main wheat species, as Maghrebi farmers later called DW simply gamh (wheat) [27]. At the end of

the Roman Empire, BW (siligo) started to penetrate from the north, to become the most important wheat species in the Italic Peninsula, with DW being displaced and restricted to the south [17,63]. During the 19th century, Italy had a stable (although slightly and steadily increasing) DW area ranging from 1.0 to 1.5 million ha, which represented around 25% of total wheat area. This proportion increased to about 70% in the late 20th century and beginning of the 21st century. The traditional regions of DW cultivation were in the south (Mezzogiorno), especially in Sicily, with more than half of the Italian acreage for many seasons, followed by Sardinia, Puglia, and Basilicata [64]. In recent years, part of its cultivation has moved to the center of the country and even to some areas of the north (Toscana, Marche, Emilia Romagna, etc.). DW is mainly used for pasta production, such as spaghetti and macaroni, a trademark of the Italian cuisine, and for which the country is the world biggest producer. For this reason, Italy needs to import significant quantities of DW grain from other countries, especially Canada and USA. Landraces such as Tumminia or Timilia, Realforte, Russello, Scorsonera, and Saragolla were examples of Italian diversity of the DW germplasm [65,66]. Relevant and innovative DW breeding programs were established, starting in 1915 with the release of the cultivar Senatore Cappelli, obtained in Puglia through genealogical selection within the North African landrace Jenah Khertifa by Nazareno Strampelli from the Stazione Sperimentale di Granicoltura (Rieti). Other popular cultivars were Capeiti 8, Appulo, Trinakria, Creso, Simeto, Duilio, and Svevo [54]. Cultivar Capeiti 8 was released at the Stazione Sperimentale di Granicoltura per la Sicilia (Sicily) in 1955 by crossing Senatore Cappelli and the *syriacum* DW type (shorter and earlier maturing) Eiti 6. In 1974, the cultivar Creso was released. It came from a cross between a mutant of Senatore Cappelli (Cp B144=Castelfusano) and a dwarf CIMMYT line. The cultivar Simeto was released in 1988 by the same Sicilian research station that bred Capeiti 8. This cultivar had an excellent adaptability across different environments and high yield. It has been cultivated in a large acreage in many countries of the Mediterranean basin for more than 20 years. The cultivar Svevo, released by Società Produttori Sementi (Bologna) in 1996, stands out by the high pasta-making quality, and it is cultivated exclusively for the pasta company Barilla [54,67]. A high-quality reference genome sequence of the *cv.* Svevo was generated in 2017 by an international consortium [68].

4.2.4. Algeria

Algeria is the largest country in Africa with more than 2 million km$^2$. Most DW cultivation is in the northern side of the country (sublittoral areas), facing the Mediterranean Basin, and with a Mediterranean climate characterized by irregular rainfall ranging from 300 to 500 mm, and relatively mild temperatures compared to the south desert areas. DW has been the main wheat species in the region since before Roman times. DW acreage has historically (19th and 20th century) ranged from 1 to 1.3 Mha, about 80–90% of the total wheat area, placing the country in the top five for DW area in most of the periods considered [69]. In the mid-19th century, Algeria went from a province of the Ottoman Empire to a French colony. Landraces such as Kahla, Hamra, Adjini, and Mahmoudi were among the many cultivated in Algeria, and most of them had a West Mediterranean origin [70,71]. French colonists brought spring BW landraces from southern France (such as Touzelle), and the Balearic Islands (landrace Mahon) for making French breads. After the independence in 1962, a growing and increasingly urbanized population demanded increasing amounts of cereal grain. National grain consumption (210 kg/person/year) is among the highest in the world, with many food products, such as couscous and local bread being made out of DW [69]. Algeria is currently still one of the largest importers of DW globally. INRAA (Institut National de la Recherche Agronomique d'Algérie) is the institution in charge of agricultural research in Algeria. It is distributed in several stations (such as the El Harrach experimental station) and took over DW breeding activities: landrace selection (Bidi 17 and Oued Zenati 368), crossing programs, and evaluating and releasing new lines from CIMMYT and ICARDA [69].

### 4.2.5. India

Wheat has been an essential crop in India since antiquity. The Mohenjo-Daro and Harappa archaeological site has shown that wheat was cultivated in that region about 5000 years ago [72]. Currently, India is the world largest producer of wheat (a production of 74.25 Mt in an area of 27.2 Mha) [6]. Many landraces and wheat species are cultivated in the country, such as BW, DW, emmer, and the 'national' Indian dwarf wheat (*Triticum sphaerococcum*) [11]. DW cultivation is very old in the country, and certainly the Arabs contributed to the extension of the crop [12,72]. It was suitable for drought conditions and grown in different regions, including Punjab, Madhya Pradesh, Karnataka, Gujarat, and Bengal [73]. After the independence of the country in 1947, DW cultivation almost disappeared in northern India, due to their high susceptibility to rusts, high lodging, and low response to fertilizers, compared to BW. However, their cultivation continued in central India due to its good tolerance to abiotic stresses, such as heat and drought. Traditional dishes made with DW semolina (suji) include laddoos (sweet balls) and chapati (flat bread). Many DW wheat landraces were mixtures of several landraces, or even with other wheat species, forming populations with particular grain characteristics, such red or amber color. Bansi was a popular landrace grown in Madhya Pradesh and Karnataka, while Jalalia (amber type) was mainly present in Madhya Pradesh, and Gangakali in Bengal. Several selection programs started in the 1930s from landraces such as Bansi. One of the typical features of the Indian DW landraces was their susceptibility to rusts, especially stem rust. Therefore, researchers started to introduce foreign resistant varieties such as Gaza. Indian emmer wheat landraces (e.g., Khapli) were also used in crosses with DW to introduce resistance to stem rust. In the 1970s, the first DW dwarf lines from CIMMYT entered India with a resounding success, about 10 years after the arrival of the first BW lines from the same center. BW cultivars such as Kalyansona and Sonalika were high yielding due to a high response to fertilizers and water, and soon they were sown over a large area. Since the 1980s, DW has slightly recovered in some northern regions, such as Punjab, due to its higher resistance to Karnal bunt and the release of new high-yielding cultivars, such as PBW 34, obtained at the Punjab Agricultural University (PAU) in 1982. From 2000, the area under DW cultivation is said to be around 9% of total wheat area [11], an overestimation, since the current acreage is 0.8–1.0 Mha (with a slight increasing tendency), which represents around 3–4% [1]. The main producing areas are in the central zone, especially in the states of Madhya Pradesh, Gujarat, Punjab, south Rajasthan, and Maharashtra [74].

### 4.2.6. Morocco

DW was the main historical wheat species in the country. The Romans extended the cultivated DW area along the sublittoral part of the north of Morocco, and Arabs reinforced the cultivation of this crop, bringing new landraces from the Mediterranean Levant. Interestingly, as in other countries of the Maghreb, BW wheat was restricted to oases and some mixtures present in DW fields called 'mule tail'. Zrea, Trikia, Maizza, and Asker were the most common DW landraces sown by farmers [26]. Ground grains of these cultivars were used to make couscous, the national dish. DW area was about 95% of all wheat area, which amounted to 0.4–0.6 Mha during the 19th and beginning of the 20th century. French colonists started to cultivate BW in 1912, and this crop rapidly gained ground. This expansion helped to supply grain for internal consumption, and to export grain to France [75]. By the late 1940s, DW area was around 1.0 Mha, while bread wheat acreage ranged 0.3–0.4 Mha [20]. By this time, a growing population demanded products rich in carbohydrates. Authorities supported BW cultivation in the 1980s, and BW products became an important part of the diet, especially in urban areas [26]. By 1986, both wheat species had similar acreages of a bit more than 1.0 Mha, but the production was not enough to supply to the population, and grain importation was needed to ensure food security. Currently, DW occupied around 35% of all wheat area, which amounts to approximately 0.9 Mha. DW is mostly sown in different areas of the country, but the humid and sub-humid areas of the center and north, with annual precipitations of about 450 mm,

permits fair yields in rainfed conditions. In the 1980s, the cv. Karim became popular. This cv. is a CIMMYT line derived from the cross Cocorit 71/Jori 69, selected as an outstanding line in Morocco, with the first batch of commercial seed coming from Tunisia, where the same line was also released. Karim kept being broadly cultivated in the 1990s and 2000s, and it is still sown by Moroccan farmers to a significant extent. Marzak and Acsad-65 were also popular cultivars in the 1980s and 1990s. New cultivars obtained in the country are Louiza, Faraj, Nassiva, Chaoui, Amria, and Marouane. DW breeding programs are assisted by CIMMYT and ICARDA. DW is a crop with a future in the country, but the competition with BW, the DW price and availability in the international market, and the threat of climate change (with predictions of a 25% drop of precipitations in dry regions) are variables that will influence the cultivation of this crop in the future.

### 4.2.7. Syria

DW has always been an important crop in Syria. The existence of wild emmer (the ancestor of cultivated tetraploid wheat species) in many parts of the country and the presence of cultivated emmer in several archaeological sites (such as Abu Hureira and Tell Aswad) have led researchers to think that DW originated in the country, or at least in the region [76]. Wheat is grown under rainfed conditions in the north, while, in the Euphrates valley in the south, it is mainly irrigated. The most important landrace in the country is Hourani, or better, the Houranis, since there are several kinds (such as Nawawi, and Auobeih). Other important landraces were Auobeih, Chehani, and Chahbaa. Most landraces belong to the *syriacum* type, whose plants, spikes, and awns are shorter, earlier, have more erect leaves, and show adaptation to drought and salinity [77]. The DW percentage during the 19th century was about 75%, which accounted for 0.6–0.8 Mha. In the first half of the 20th century, the acreage increased to 1 Mha. Senatore Cappelli was introduced in 1932. In 1964, a fruitful collaboration between Syrian authorities and CIMMYT was started and led to new material testing and training of young Syrian researchers. In 1968, the Arab Center for the Studies of Arid Zones and Dry Lands (ACSAD) was established in Damascus and contributed to advanced germplasm transfer and exchange of researchers for other Arab countries. The international center ICARDA, located at Tel Hadya, Alepo, was established in 1977. It was important for agricultural research, exchange of genetic material, and training of national staff [78]. In the period 1990–2017, DW acreage was 0.8–1.0 Mha (about 50–70% of the total wheat area), but in the last seasons has dropped to about 0.5 Mha (37%). At ICARDA, Hourani was crossed with the high yield Mexican cultivar Jori 69, to obtain Cham 5 (also called Om Rabiâa), released in 1994 with considerable impact in several countries. Douma 1 and Douma 3 are examples of other cultivars obtained and released in the country. As part of the wheat in Syria is irrigated, one of the main breeding objectives is to get salt-tolerant cultivars. Gamma radiation of Hourani (already with a certain level of salt tolerance) of 15 Krad were carried out to obtain several promising tolerant lines [78].

### 4.2.8. Tunisia

Tunisia is the most northern African country. As a WANA country, the dependence of wheat products, especially DW, is very high. Although the country has several agroclimatic zones, wheat is grown primarily in the north of the country, under rainfed conditions. Drought and Septoria tritici blotch are the main constraints for wheat production in the country. DW was, by far, the main wheat species during the 19th century with acreages ranging 0.2–0.4 Mha. BW was only found in mixtures with some DW landraces [27]. From the French protectorate era (1880s), BW started to be cultivated, but always as a minor crop, with an acreage lower than 10% of total wheat. In 1910–1950, DW was about 0.5–0.6 Mha (95% of total wheat). Starting in 1951, DW acreage increased to about 0.7–0.8 Mha and remained around that level until 2000. Since then, DW area has slowly decreased to 0.4–0.5 Mha. In Tunisia, DW has withstood its ground in relation to BW and remain by far the major cereal crop, representing at least 70–80% of total wheat. The main DW landraces grown until the 1970s were Derbessi, Agini, Agili, Mekki, Hamira, and Jenah Khotifa. In

the 1970s and 1980s, new cultivars from CIMMYT were introduced and rapidly displaced the old landraces with great impact on the national production and productivity. These included cultivars such as Ben Bechir, registered in 1980, but the most popular being Karim 80, rapidly reaching over 60% of the national DW area. It is still a major cultivar grown by farmers 30 years after its releasing. Razzak 87 and Khiar 92 were the next two promising cultivars. They have a fair and stable yield but were susceptible to Septoria tritici blotch. Nasr 2004, selected from an ICARDA cross, was the first cultivar released for its acceptable resistance to Septoria tritici blotch [79]. However, the more recently released cultivar Maali, from a local cross and using local selection, is the only one that has finally replaced Karim 80 and is becoming the major cultivar in the country.

### 4.2.9. USA

The USA, a young country without tradition of DW cultivation, appears as one of the historically important DW-growing countries. No official data were found until 1920 [80], but since that year the percentage wheat area sown to DW has been about 5%. The first records of DW in the USA date back to 1854, and, in 1864, the landrace Arnautka (also called Nicaragua) was introduced by the USDA [47]. At that time, DW was mainly sown in Texas (where Mexican landraces were also grown) and widely used for feed [81]. The real breakthrough for the crop came in 1900 when M.A. Carleton promoted the cultivation of the Russian landrace Kubanka [47]. This variety became the leading cultivar from 1910 to 1920, and it is in the pedigree of many modern US cultivars [47]. DW production increased markedly thereafter to start the USA pasta industry. The US stem rust outbreak of 1904 did not affect much DW as it did BW [47]. In a few years, cultivation of this crop shifted northward to become a spring crop in North Dakota, which would be the main DW producing state of the country to date. According to the estimates of this study, by 1910, the country was the sixth in the world in terms of DW acreage (0.88 Mha), and fourth in 1920 (1.61 Mha). By 1930, it rose to the third place (1.87 Mha), just after Russia and India, and surpassing Turkey, a large all-time DW producer. From the 1930s to the 1970s, the acreage decreased to approximately 1 Mha. In the 1980s, the area increased again to over 2 Mha. It coincided with an increase of global pasta demand. The record seasons are 1981/1982 and 1982/1983, with about 2.3 Mha [80], earning the US the first place in DW acreage in the world. From 1998–2004, the area stayed over 1 Mha. Then, it started to slowly decrease, and currently ranges between 0.5 and 0.8 Mha. The US DW acreage is characterized, during all periods, by high fluctuations between seasons. Currently, along with North Dakota, DW is also produced in other states of the north (Montana), and the south, such as Arizona and California [82]. The first breeding efforts started in the form of line selections from the cultivar Kubanka, producing the cultivar Acme in 1909. At the North Dakota Agricultural Experiment Station, a national DW breeding program was initiated in 1929 that produced cultivars Stewart (1943) and Langdon (1955) [82,83]. The main current program in the USA is led by the North Dakota State University (NDSU). Grain yield, disease resistance, and quality traits are the main objectives of the breeding program. Among the main obtained cultivars are Maier (1998), Plaza (1999), Dilse (2002), and Alkabo (2005) [82].

### 4.2.10. Spain

The Roman Empire first and the Arab Empire later brought DW cultivation and many landraces to the country [84]. In 1800, there were 2.9 million ha of wheat [18], and DW occupied approximately 18% of total wheat area (about 0.5 Mha), most of it in the south (Andalusia), but also in the east (Murcia, Valencia, Balearic Islands), reflecting the Mediterranean character of the crop. The main use of DW was bread (with semolina alone or mixing with BW flour), although a kind of short thin noodles (fideos) were a common food all over the country [85]. Fanfarrón, Recio, Rubio, or Raspinegro were the names of some Spanish landraces. Most of them have a clear west Mediterranean origin, especially from the Maghreb and Sicily, with names such as Moruno (Morish) or Siciliano (Sicilian) [17,70]. Wheat acreage grew during the 19th century to over 4.5 million ha, and

with it that of DW, to some 0.6–0.8 Mha. However, after 1950, DW acreage fell to 0.25 Mha in 1964, and to a minimum of 0.09 Mha in 1978 [19]. DW vanished from the east and southeast, and, for a period, the crop survived only in west Andalusia (Seville and surrounding provinces). Many BW cultivars were introduced in the 1940s and 1950s of Italian (Mara, Impeto) and French origin (Florence Aurore), occupying much land previously devoted to DW. From the early 1970s, a new wave of spring semidwarf BW cultivars from CIMMYT (Cajeme, Siete Cerros, Yécora) had the same effect. During the 1990s, DW recovered due to an increase of pasta demand (national and global) and to European Union subsidies for traditional DW growing areas [56]. Acreage rose again to 0.8–0.9 Mha in 1999–2005, but when the subsidies were withdrawn in 2007, it fell but stabilized around 0.3–0.4 Mha, a bit more than the current area (70% in Andalusia, 25% in Zaragoza, 5% in other regions) [85]. The Italian Senatore Cappelli was cultivated between the 1930s and 1970s, and in the Agricultural Research Farm of Jerez de la Frontera, J.B. Camacho crossed Senatore Cappelli with some Spanish landraces to obtain cultivars Ledesma and Andalucía 344, quite popular in the 1950s and 1960s [86]. In the late 1970s and 1980s, new DW cultivars from CIMMYT (e.g., Cocorit 71, Mexa, and Yavaros C79) arrived in Spain, displacing much of the local landraces. Italian cultivars, such as Simeto, were also cultivated by Spanish farmers in the 1990s and 2000s. There have been public breeding programs, but a lack of continuity of those programs was frequent. The DW program at IRTA (Lleida) yielded cv. Euroduro, while the national program of INIA led to Hispasano. Nowadays, the most grown cultivars come from private companies like Limagrain (Athoris), Agrovegetal (Don Ricardo), and Guadalsem (Amílcar), most of them derived from CIMMYT advanced lines. In addition, the National Center of Plant Genetic Resources (CRF) preserved more than 403 accessions of Spanish landraces, and the collection has been phenotyped and genotyped in several studies [87,88].

4.2.11. Ethiopia

In Ethiopia, DW has been cultivated for centuries and an extraordinary diversity and uniqueness of landraces has been reported in many works, such as the one described by Nicolái Vavílov in his travel to this country in the 1920s [89]. Kabbaj et al. [58] demonstrated that Ethiopian DW landraces clustered separately from landraces derived from other areas of the world, such as the ones from the rims of the Mediterranean and Black Sea, so much so that Ethiopia is currently considered a secondary center of origin of this crop. There are not clear data before the 1960s, but all information available indicates that DW occupied 60–70% of the total wheat area until 1990, when BW (as in other countries) started to gain ground [45]. In the 1960s and 1970s, DW area was estimated in around 0.6 Mha. Thereafter, it started to decrease. The wide adaptation of BW material from the CIMMYT program displaced much of the acreage previously devoted to DW, and currently only about 15% of total wheat area is planted with this crop (about 0.27 Mha). Additionally, new DW cultivars are not easily reached by farmers, due to a problematic seed sector, the size of the country, and the low purchasing capacity of farmers. About 80% of the DW area was sown with landraces still in 2015 [32], such as Aybo, Set-Akuri, Arendeto, Loko, Kurkure, and Mengesha [90]. At tropical latitudes, DW is sown at altitudes between 1800–2700 m in the highlands black clay soils. Sowing is usually performed in August, just after the rainy season, and harvest time takes place in a dry December. Although currently the main product of DW is pasta, local bread (*injera*) was and is made with the flour of this wheat species. Wheat straw is also used as animal feed and as roof thatching material [91]. At present, the Ethiopian Biodiversity Institute (EBI) hold a collection of more than 7000 landraces [91], and the National Durum Wheat Breeding Program is located at the Debre Zeit Agricultural Research Center. The main goals of the breeding program are raising yields, lodging resistance, resistance to leaf and stem rust (Ug99 lineage), and drought and heat tolerance. In addition, a good network of multilocation yield tests is being developed to target specific adapted genotypes or to select genotypes with wide adaptation and a stable yield across the different agroecological regions of the country [45].

### 4.2.12. Other *T. turgidum* Subspecies

Other subspecies of *T. turgidum* appeared in the historical wheat literature along with DW, particularly rivet and emmer wheat. Rivet (or poulard) wheat (*T. turgidum* ssp. *turgidum*) is a close subspecies of DW, and according to some studies, genetically indistinguishable [92], although morphologically several differences can be detected (taller plants, glaucous leaves, later ripening, thicker stems, etc.). It was estimated that some 0.14 Mha (140,000 ha) were sown in Spain in 1898, and 110,000 ha in 1935 to these DW relative (Martínez-Moreno, unpublished data). In Italy, 70,000 ha were cultivated to this subspecies in 1927 [64]. Ethiopia, France, Greece, the Balkans, Portugal, Russia, and Turkey had also a relative important acreage of rivet wheat [81,93]. The importance of this wheat decreased during the 20th century to almost null by 1970. Something similar occurred with emmer (*T. turgidum* ssp. *dicoccum*). During the Roman period, it was mostly replaced by DW [17], but, in the 19th century, it was still of some importance in several countries, being considered more tolerant to biotic (e.g., to stem rust) and abiotic stresses (that permitted its cultivation in mountainous areas), and with a hulled grain adequate for feed. In Turkey, there were still 56,000 ha by 1927, but only 13,000 ha in 1993 [94]. In Ethiopia, 42,700 ha were still sown in 1990–1996, and India had 50,000 ha in 2000 [12]. The small acreage of these two sister subspecies of DW does not interfere with the results obtained in this study, since the percentage of DW was estimated from total wheat acreage, which included all wheat species.

### 4.2.13. Global DW Area in the Future

It seems that DW will be cultivated approximately in the same area as today, for producing pasta, couscous, and many other products that can be made with the DW grain. The future acreage of the crop will depend on several factors, including the performance of new cultivars and their competitivity with bread wheat in different environments, their ability to withstand biotic and abiotic challenges in the future, and the existence of a price differential favoring DW over BW. The possibility of breeding hard winter BW cultivars for producing pasta in cold countries (Germany, Hungary, some regions of the USA, etc.) may slightly reduce the DW area [49]. In warmer countries of the WANA region (the core area of this crop), DW area may also decrease in favor of BW, but not so in Mediterranean European countries, since DW is considered a cereal for making premium products, such as pasta. If DW grain prices are still as high as today (around 500 €/t in 2021/22), countries with extensive cereal land such as Argentina, Australia, Russia, or Kazakhstan may increase the area of this crop. In addition, climate change must be taken also into account. In a recent study, it was estimated that global warming may reduce the world suitable potential area for DW cultivation by 19% in 2050 and by 48% in 2100. Mediterranean regions and North America accounted for the most suitable land losses. However, new opportunities for the crop may be open in central and western Europe, and Russia [95].

## 5. Conclusions

In this article, an approximation to the historical area sown with durum wheat from 1800 to the present was provided. It was possible to monitor and discuss the evolution of the area of this crop globally and in the countries with the largest historical area. Some of these countries have preserved the cultivation of this crop (e.g., Tunisia, Algeria), others have partially lost it (e.g., Russia, India), others have increased the DW area (e.g., Italy, Greece), and some countries have incorporated this crop into their current agriculture (e.g., Canada, France). Competition with BW and the appreciation of the differentiated products to which DW gives rise (e.g., pasta and couscous) are key factors to understand the dynamics of this crop in the past, present, and future. The data from this study may help to better understand the evolution of wheat diseases with races specialized on DW (Septoria tritici blotch, leaf rust, stem rust, yellow rust, common bunt, etc.), changes in flour quality and culinary traditions in particular countries, and changes in agriculture practices in the last two centuries.

**Supplementary Materials:** The following supporting information can be downloaded at: https://www.mdpi.com/article/10.3390/agronomy12051135/s1, Video file; Figure S1: Animated bar chart of the DW acreage evolution of the 15 main historically cultivating countries in 1800–2020; Spreadsheet files; Table S1: Wheat historical area in 37 important countries regarding cultivation of this crop, and world wheat area (Mha) in several seasons of the period 1800–2020; Table S2: Durum wheat area of the main historically durum wheat-growing countries (24) and world durum wheat area (Mha) in the period 1800–2020; Table S3: Percentage of durum wheat to all wheat area (%) of the main historically durum wheat-growing countries (24) in the period 1800–2020.

**Author Contributions:** Conceptualization, F.M.-M.; methodology, F.M.-M.; validation, F.M.-M., K.A. and I.S.; formal analysis, F.M.-M.; investigation, F.M.-M.; resources, F.M.-M. and I.S.; data curation, F.M.-M. and K.A.; writing—original draft preparation, F.M.-M.; writing—review and editing, F.M.-M., K.A. and I.S.; visualization, F.M.-M.; supervision, F.M.-M., K.A. and I.S.; funding acquisition, F.M.-M. and I.S. All authors have read and agreed to the published version of the manuscript.

**Funding:** This research was funded by FIUS (Research Foundation of the University of Seville), grant number PRJ202104359.

**Institutional Review Board Statement:** Not applicable.

**Informed Consent Statement:** Not applicable.

**Data Availability Statement:** Not applicable.

**Acknowledgments:** Many thanks are due to the International Grains Council (IGC) Secretariat for the provided information in recent years. The assistance provided by Antonio Blanco (University of Bari, Italy), Nasserlehaq Nsarellah (INRA, Settat, Morocco), and Irfan Özberk (University of Harran, Turkey) was greatly appreciated.

**Conflicts of Interest:** The authors declare no conflict of interest. The funders had no role in the design of the study; in the collection, analyses, or interpretation of data; in the writing of the manuscript, or in the decision to publish the results.

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
