# Peer review of "Global Changes in Cultivated Area and Breeding Activities of Durum Wheat from 1800 to Date: A Historical Review"

_agronomy, doi:10.3390/agronomy12051135_

Round 1
Reviewer 1 Report
I have reviewed a previous version of the MS and my decision back then was to accept it after I had reviewed it's first version. The MS is very interesting, very important and well presented. Coming from a country where durum wheat is essential for food security I understand well the importance of this study and its singificance. My decision is still to accept it and to congratulate the Authors for their important efforts.
Author Response
Reviewer 1
I have reviewed a previous version of the MS and my decision back then was to accept it after I had reviewed its first version. The MS is very interesting, very important and well presented. Coming from a country where durum wheat is essential for food security I understand well the importance of this study and its significance. My decision is still to accept it and to congratulate the Authors for their important efforts.
Thank you so much.
Reviewer 2 Report
This manuscript aims to review the history of durum wheat cultivation and breeding from 1800 to 2020 in the main productive countries in the world. The data are quite informed and their presentation are well structured. Much work has been done in historical evolution especially in the Mediterranean basin. However, the information presented here is sound and of great importance in terms of the evolution of durum wheat, the comparison of durum and bread wheat as well as the future breeding programs to face climate changes.
The paper is within the subject of this Special Issue, and it can be accepted by the “Agronomy” after minor revision, according to the below comments:
- Abbreviations should be inserted in the text in order to facilitate the reader,
- Table S3 should be referred in the text in Materials and Methods,
- Relative references have to be inserted like:
- Christopoulos and G. Ouzounidou 2020. Climate change leading to post harvest losses in breadwheat. In Climate change and food security with emphasis on wheat eds M. Ozturk and Alvina Gul, Elsevier books, pp257-264.
- Ouzounidou, M. Moustakas and E. P. Eleftheriou. 1997. Physiological and ultrastructural effect of cadmium on wheat (Triticum aestivum L.) leaves. Archives of Environmental Contamination and Toxicology 32: 154-160.
- A re-arrangement of the Conclusions must be made, starting from the conclusions of their data presentation and after their suggestions for breeding programs.
Author Response
Reviewer 2
This manuscript aims to review the history of durum wheat cultivation and breeding from 1800 to 2020 in the main productive countries in the world. The data are quite informed and their presentation are well structured. Much work has been done in historical evolution especially in the Mediterranean basin. However, the information presented here is sound and of great importance in terms of the evolution of durum wheat, the comparison of durum and bread wheat as well as the future breeding programs to face climate changes.
The paper is within the subject of this Special Issue, and it can be accepted by the “Agronomy” after minor revision, according to the below comments:
- Abbreviations should be inserted in the text in order to facilitate the reader,
Abbreviations of durum wheat (DW) and bread wheat (BW) appear the first time these words are mentioned in the text. I realized I wrote the abbreviation of bread wheat the second time it was mentioned. This warning helped me to the correct this, thanks. Alternatively, I can make a list of abbreviations, but I think it is not necessary.
- Table S3 should be referred in the text in Materials and Methods,
Ok, I have added it in the new version.
- Relative references have to be inserted like:
- Christopoulos and G. Ouzounidou 2020. Climate change leading to post harvest losses in breadwheat. In Climate change and food security with emphasis on wheat eds M. Ozturk and Alvina Gul, Elsevier books, pp257-264.
- Ouzounidou, M. Moustakas and E. P. Eleftheriou. 1997. Physiological and ultrastructural effect of cadmium on wheat (Triticum aestivum L.) leaves. Archives of Environmental Contamination and Toxicology 32: 154-160.
- A re-arrangement of the Conclusions must be made, starting from the conclusions of their data presentation and after their suggestions for breeding programs.
I have formatted the references according to the style of Agronomy (https://www.mdpi.com/journal/agronomy/instructions). I have used Mendeley and in the books I am not able to add the number of pages. I am going to ask for assistance to the editor.
Reviewer 3 Report
Dear Authors,
This paper is scientific-relevant, but the reading it is very difficult.
Therefore, I suggest that:
- each mentioned country has to be enumerated as subsections (4.2, 4.3, etc.)
- in all initial paragraph some introduction needs to be involved for each country.
- Also, English needs to be improve for better and easier reading of the paper.
- involve Figure in the main text
- do not use "we" and "our" - use a passive form for all sentences
- in the conclusion part avoid using references, you have the opportunity to use them in the main part of the text. I suggest reorganizing the conclusion part.
Author Response
Reviewer 3
Dear Authors, This paper is scientific-relevant, but the reading it is very difficult. Therefore, I suggest that:
- each mentioned country has to be enumerated as subsections (4.2, 4.3, etc.)
Ok, I wrote the sub-subsection of each country within the subsection 4.2. (cultivation and breeding in the main DW-growing countries…).
- in all initial paragraph some introduction needs to be involved for each country.
I tried to write an introduction for each country in each paragraph. However, to keep this review article relatively short (if 11,340 is short), I only wrote a brief introduction for each country.
- Also, English needs to be improve for better and easier reading of the paper.
Ok, me and the co-authors are going to read the article once more to detect errors and improve the English of the article.
- involve Figure in the main text
Ok, already done.
- do not use "we" and "our" - use a passive form for all sentences
Ok. I have already changed the sentences containing ‘we’ and ‘our’ to a passive style.
- in the conclusion part avoid using references, you have the opportunity to use them in the main part of the text. I suggest reorganizing the conclusion part.
I tried to reorganize the conclusions section.
Best
Reviewer 4 Report
The paper thoroughly describes global and regional changes in cultivating area of durum wheat, as well as regional durum lanraces. I believe that this article, along with supplementary material, is a reach overview on durum wheat history. I recommend publishing it in the journal Agronomy.
Below some minor comments:
- Abstract: I suggest adding information about which country is now the biggest durum cultivator (in the last part of this paragraph).
- Line 57-58: You referenced twice the same materials. I don’t know if it was intentional, because you wanted to introduce names of those authors, or it was a simple mistake. If it was the latter, than please remove citations “(Royo et al., 2005) “ and “(Bonjean and Angus, 2001; Bonjean et al., 2011, 2016)”.
- Table 1: Please put the percentage symbol below caption ‘Durum wheat area’. Then all units (Mha and %) will be listed below columns’ captions.
- Figure 1: I suggest slightly changing the caption, e.g.: “Historical durum wheat area in the nine main durum cultivating countries in 1800-2020.”
Also, could you use more diverse colours to differentiate countries (especially blue ones)?
Why have you omitted Canada in both figures? It has now the biggest durum area.
- Figure 2: I suggest slightly changing the caption, e.g.: “Historical durum wheat area (%) to all wheat area in the nine main durum wheat-growing countries in 1800-2020.”
Also, as in fig. 1, could you use more diverse colours to differentiate countries (especially blue ones)?
I suggest changing the description of vertical axis to “Durum wheat area”, because the axis unit is %, not Mha.
- Line 161: Please change the word ‘is’ to ‘was’.
- Line 222: Please use symbol ° (now it’s underlined).
- Line 286: Please add a space: ‘1.1and’.
- Line 540: Please change ‘gained’ to ‘gain’ and ‘were’ to ‘was’.
- Line 546: Please change ‘are’ to ‘was’.
- Lines 554-556: Please reshape this sentence: “At present, the Ethiopian Biodiversity Institute (EBI) hold a collection of more than 7,000 landraces [77], and the National Durum Wheat Breeding Program is located at the Debre Zeit Agricultural Research Center, plays a centralized role in DW research and development.”
Did you mean that both the EBI and the Debre Zeit Agricultural Research Center play a centralized role in DW research and development? Or only the Debre Zeit Agricultural Research Center?
Author Response
Reviewer 4
The paper thoroughly describes global and regional changes in cultivating area of durum wheat, as well as regional durum landraces. I believe that this article, along with supplementary material, is a reach overview on durum wheat history. I recommend publishing it in the journal Agronomy.
Thank you so much.
Below some minor comments:
- Abstract: I suggest adding information about which country is now the biggest durum cultivator (in the last part of this paragraph).
Ok, done it.
- Line 57-58: You referenced twice the same materials. I don’t know if it was intentional, because you wanted to introduce names of those authors, or it was a simple mistake. If it was the latter, than please remove citations “(Royo et al., 2005) “ and “(Bonjean and Angus, 2001; Bonjean et al., 2011, 2016)”.
Yes, it was intentional because these three books have been instrumental to carry out this study.
- Table 1: Please put the percentage symbol below caption ‘Durum wheat area’. Then all units (Mha and %) will be listed below columns’ captions.
Yes, now this Table looks better.
- Figure 1: I suggest slightly changing the caption, e.g.: “Historical durum wheat area in the nine main durum cultivating countries in 1800-2020.”
Ok, done it.
Also, could you use more diverse colours to differentiate countries (especially blue ones)?
I have changed colors. Now there are more contrast between colors.
Why have you omitted Canada in both figures? It has now the biggest durum area.
Yes, but Canada has not been an historical player regarding durum wheat cultivation. Just in 1950, durum wheat acreage reached 300,000 ha, but even in the 1940s barely reached 100,000. It is a relative newcomer. However, I created a video (supplementary, Figure S1) of an animated bar chart of the durum acreage evolution of the 15 main historically cultivating countries in 1800-2020 and Canada is in it.
- Figure 2: I suggest slightly changing the caption, e.g.: “Historical durum wheat area (%) to all wheat area in the nine main durum wheat-growing countries in 1800-2020.”
Ok, done it.
Also, as in fig. 1, could you use more diverse colours to differentiate countries (especially blue ones)?
I have changed colors. Now there are more contrast between colors.
I suggest changing the description of vertical axis to “Durum wheat area”, because the axis unit is %, not Mha.
Ok, in fact, I deleted legends to give more space to the graphs.
- Line 161: Please change the word ‘is’ to ‘was’.
Ok, done it.
- Line 222: Please use symbol ° (now it’s underlined).
Ok, done it.
- Line 286: Please add a space: ‘1.1and’.
Ok, done it.
- Line 540: Please change ‘gained’ to ‘gain’ and ‘were’ to ‘was’.
Ok, done it.
- Line 546: Please change ‘are’ to ‘was’.
Ok, done it.
- Lines 554-556: Please reshape this sentence: “At present, the Ethiopian Biodiversity Institute (EBI) hold a collection of more than 7,000 landraces [77], and the National Durum Wheat Breeding Program is located at the Debre Zeit Agricultural Research Center, plays a centralized role in DW research and development.” Did you mean that both the EBI and the Debre Zeit Agricultural Research Center play a centralized role in DW research and development? Or only the Debre Zeit Agricultural Research Center?
I have rephrased and shortened the sentence as follows: ‘At present, the Ethiopian Biodiversity Institute (EBI) hold a collection of more than 7,000 landraces [91], and the National Durum Wheat Breeding Program is located at the Debre Zeit Agricultural Research Center.’
Only Debre Zeit ARC plays a role in DW research.
Best
This manuscript is a resubmission of an earlier submission. The following is a list of the peer review reports and author responses from that submission.
Round 1
Reviewer 1 Report
The manuscript presents a very interesting contribution to the field. The objectives are well listed as is the scope of the study. However, there are some elements that need to be addressed. In addition, a review of English is recommended as their are some flaws.
Abstract
a. Line 10: 2020 and 2021 are in the past please use the past tense in this sentence.
b. Line 12: a historical estimation not an.
Introduction
a. The Mediterranean basin is more propertly suited that the rim (line 30)
b. Line 33: Being the most important exporters I assume?
c. Lines 35-36: The mentioned elements are also main components of the MENA region's dietary regime aswell, please add that.
d. Line 52: When they state not states
e. Line 55: most important DW cultivating countries is more linguistically correct.
f. The Authors should include some references of previous works on the topic. Surely, there are other works that have attempted to do the same. These should be listed as sufficient background information on the study is lacking.
2. Materials and methods
a. Line 65: instead of referred, please use listed.
b. Line 66: How did the Authors find that the DW acreage is overestimated? Please explain or add a supporting citation
c. Line 68: from different countries
d. The Authors have explained and accounted well for the changes of geographical borders. A strong point that should be underlined, congratulations on that.
e. It could be interesting to have a listed ranking of countries following the Tables 1-3
3. Results
a. This table (line 144) is not Table 1, it should be Table 4. Please check the numbering.
b. Line 183: Most not more
c. The historical section and the historical narrative per country is a very strong point. Congratulations
d. Line 326: irregular rainfall instead of moody
e. Line 490: no official data not "not"
f. Why is the situation of the USA unexpected? Could the Authors provide an explanation?
g. The USA is not entirely a cold country, please use in some regions of the USA
5. Conclusion
First, please pay attention to the numbering it should be 4 not 5. Please develop this section further, the article cannot have a conclusion of 5 lines only. Please add some more findings and perspectives. Maybe merging section 3.2. could help
Reviewer 2 Report
This paper extensively describes the historical development of durum wheat, especially compared to bread wheat, in the main producing countries. This is done in a very detailed manner. The only thing that I miss here it the why question. There is no storyline that convinces the reader. The paper could make an excellent contribution to a book about wheat, but from a scientific perspective, it lacks a main message. However, there is enough information in the paper allowing to find a message. Especially given the present issues the world faces, e.g. regarding the dramatic situation in one of its bread baskets today, i.e. the situation in Ukraine.
To improve the paper, I suggest to move unnecessary detail to a Supporting Information section, so that a main story remains.
Next, I checked some data. Table 2 gives durum wheat areas, also for India and Pakistan. However, in 1800, India as such also did not exist. India, Pakistan and Bangladesh were one country. Next, after the separation of India on the one hand and Pakistan and Bangladesh on the other, we see that the durum area goes from 1.8 in 1940 to 0.294 and 0.082 in 1950, which does not match the area in 1940. Also the separation of Bangladesh is not clear.
The wheat area in Pakistan is then 8.6 Mha in 2020, while the official agricultural statistics report an area of 10 MHa. I suggest to check all the numbers.
Round 2
Reviewer 1 Report
The Authors have addressed the comments well. The manuscript is good for publication. Congratulations